# Analysis of the Stability of a Flat Textiles with a Two-Parameter Deflection Curve

**DOI:** 10.3390/ma17020503

**Published:** 2024-01-20

**Authors:** Piotr Szablewski

**Affiliations:** Department of Mechanical Engineering, Informatics and Chemistry of Polymer Materials, Lodz University of Technology, Zeromskiego 116, 90-924 Lodz, Poland; piotr.szablewski@p.lodz.pl

**Keywords:** stability analysis, large deflections, textile mechanics, elastica, state of equilibrium, numerical modeling, principle of virtual work, energetic method

## Abstract

The issue of large deflections of textiles occurs primarily when analyzing the “drape” of curtains, tablecloths and other flat textile products. The correct drape is particularly important from an aesthetic point of view. Additionally, there is a problem with the stability of the folds created during the drape process. The analysis of this problem is difficult due to the occurrence of large deflections and non-linear properties of the material. In this article, a selected fragment of the above-mentioned issue was tested, relating only to the stability of the fold formed under given loading conditions. A typical example is a fabric resting on a flat surface and loaded with compressive forces. The presented considerations lead to obtaining the deflection curve for a given self-weight and compressive force. Additionally, the obtained shape was tested for stability. Two shape parameters used in the analysis can be applied for the simulation of different shapes of the deflection curve. The analysis has been made using the energy method relating to the total potential energy of the object. The obtained results may be used in algorithms and simulation programs for fabric folding, buckling and for another applications in the area of textile mechanics.

## 1. Introduction

The analysis of the stability of mechanical structures is a research topics undertaken by many scientists. Overall, this issue is not simple and often requires complex mathematical considerations. Similarly, taking into account large structural deflections (e.g., beams, plates and shells) often leads to very complicated mathematical solutions. In textile mechanics, large deflections and stability problems are often considered.

This is particularly important when analyzing the “drape” of curtains, tablecloths, clothing, etc. Due to the fact that the mechanics of textiles is a relatively novel scientific field, many problems are solved on the basis of classical mechanics, the theory of elasticity, etc. Moreover, many researchers often use the classical theory of “elastica” for deflection analysis. The issues presented above are the main topic of the work.

The stacking or folding of the flat textile products can be analyzed using Peirce’s cantilever bending theory [1]. Hearle et al. [2] have pointed out that textile structures can be classified into an hierarchical structure representing the deformations of one-, two- and three-dimensional continua. The properties of a particular continuum depend generally on the continuum properties of lower-level units and the interconnection. These relationships have been studied for yarn structures and deformations, as well as fabric structures (Hearle et al. [3], Konopasek and Shanahan [4]), but little work appears to have been performed on the deformations of fabrics treated as two-dimensional continua, partly due to the difficulty in characterizing the elastic properties of textile sheets (Shanahan et al. [5]), and partly due to the difficulties in obtaining solutions (Lloyd [6]). An interesting article was written by Lloyd et al. [7]. In this work an example of a fabric folding under its own weight is analyzed using the bending curve theory to model the fabric cross-section. The reduction in the governing differential equations, and the boundary conditions appropriate to each stage of the motion in a dimensionless form reveals that the parameters which characterize the deformed state depend only on the fabric bending length and feeding height. The form of the function is determined by repeated numerical solutions. Lloyd [8] considered the drape process of textile structures. In this work, the special difficulties of modelling the complex deformations of textile fabrics are discussed in the context of flexible shell theory, and the particular requirements of analysis methods in this area are described.

Deflections of plates or beams are very often the subject of many engineering tasks that have applications in everyday life. These considerations can be successfully applied to the analysis of textile structures. The theory of small deflections of beams subjected to concentrated forces is the subject of many studies in the area of mechanics, e.g., the study by Gere [9]. However, in the case of small deflections, the principle of superposition applies, as well as the assumption of a small bending angle. Moreover, the obtained equations show the proportionality between deflections and the forces causing them. If the deflections are large, the assumption of a small angle and the principle of superposition do not apply. The problem becomes difficult to solve. The analytical solution does not exist due to the presence of nonlinear terms in the equations.

Lee et al. [10] analyzed the large deflections of elastic beams with various cross-sections. They assumed different loads and used the Runge–Kutta–Falsi method for analysis. Baker [11] obtained the large deflections of elastic beams under distributed loads using a weighted residual solution for the Bernoulli–Euler equation. Dado and AL-Sadder [12] presented a new analysis of the large deflections of non-prismatic beams based on the integrated least square error of the nonlinear governing differential equation in which the angle of rotation is represented by a polynomial. Shatnawi and AL-Sadder [13] studied exact large deflection of non-prismatic, nonlinear bi-modulus cantilever beams subjected to a tip moment by applying a power series approach to analytically solve highly nonlinear, simultaneous, first-order differential equations. Shvartsman [14] analyzed large deflection of beams loaded with a follower force. He reduced the nonlinear two-point boundary-value problem to an initial value problem. To do this, he changed the variables and then solved the problem without iteration. The numerical analysis of the Peirce’s test has been performed by Szablewski and Kobza [15]. For the analysis, they used a mathematical model of heavy elastica. AL-Sadder and AL-Rawi [16] developed quasi-linearization finite differences for the large-deflection analysis of non-prismatic beams subjected to various types of distributed and concentrated loads in different directions. Ibrahimbegovic [17] used finite element analysis for testing the large displacements of beams by means of the three-dimensional finite-strain Reissner beam theory.

Cantilever beams of non-linear materials have also been studied. Lewis and Monasa [18] also examined the large deflections of beams. These beams were made of non-linear materials. For the analysis, they used the fourth-order Runge–Kutta method. Lee [19] investigated the deflections of beams, made of non-linear elastic material, loaded with combined loading using Butcher’s fifth-order Runge–Kutta method. Antman [20] analyzed large buckling of nonlinear-elastic beams which were subjected to bending, torsion, tension and shear. Cesnik et al. [21] presented an improved theory of composite beams. The basis of their theory is the variational-asymptotic method. Szablewski [22] presented an original method for testing the equilibrium. In this work, the tested object was modeled as an un-stretchable elastica loaded with self-weight and axial compressive force. The large deflections of prismatic beams made of linearly elastic materials have been analyzed in numerous studies. These beams were often subjected to uniformly distributed loads. Seames and Conway [23] considered a mathematical method for finding the large deflection of a beams loaded with a distributed load. In this work, the axis of the beam is approximated by a series of circular arcs tangent to each other. Rohde [24] obtained the large deflection of a beam by expanding the arc length into a power series. Lee et al. [25] experimentally examined the displacements and stresses in deflected beams using the photoelasticity method. Szablewski and Korycki [26] presented a theory relating to determining the shape of heavy elastica subjected to bending, using the displacement method. Belendez et al. [27] considered the large deflections of uniform beams loaded with a combined load: concentrated force and uniformly distributed load. They obtained the analysis results numerically using an algorithm based on the Runge–Kutta–Felhberg method. They compared the obtained theoretical results with experimental data. Frisch Fay [28] obtained the large deflections of beams loaded with two concentrated forces using elliptic integrals. Bisshopp and Drucker [29] obtained the large deflections of beams subjected to concentrated force also using elliptic integrals. The works [28,29] were based on the classical Bernoulli–Euler theory. A similar procedure of applying Jacobi elliptic integrals of the first and second types, taking into account only the end-load, was used by Howell and Midha [30].

Cantilever beams loaded with a continuous load and a force at the free end was also analyzed by Belendez et al. [31]. They used a mixed analytical and numerical method to obtain the deflected shapes. Saxena and Kramer [32] proposed an original numerical integration method. This method requires special consideration for the existence of an inflection point in the bending beam.

Similarly, the analysis of plate stability is the subject of numerous considerations. Thick plate analysis covers research areas such as bending, buckling and vibration (Onyeka et al. [33]). Under the influence of in-plane compressive loads, the plate gradually becomes unstable at the certain value of the load. This value is the so-called critical force. We are dealing here with the phenomenon of buckling (Hassan and Kurgan [34], Onyeka et al. [35]). The phenomenon of plate buckling can be classified as elastic and inelastic buckling. This classification is based on the stress–strain relationship. The problem is considered as the elastic buckling problem if the critical load is smaller than the elastic limit of the plate material. Otherwise the problem is called inelastic buckling (Onyeka et al. [36]). The classical methods, also called the equilibrium (Euler) methods, are analytical methods that seek to obtain closed-form solutions for solving the partial differential equations of the plate-buckling problem within the plate domain, taking into account the boundary conditions of loading and the restraints of the plate edges ([Onyeka et al. [37]). To obtain approximate solutions to the plate problem, numerical methods are used. These methods include the weighted residual methods, finite difference methods, finite element methods, and finite strip methods (Ike [38]). To overcome the rigorous routine inherent in classical and numerical methods, variational method can be applied. The energy methods such as the Ritz variational method, Kantorovich variational method, Rayleigh–Ritz method, and Galerkin method, with respect to the displacement function, minimizes the total potential energy functional to derive the characteristic buckling equation from which the buckling loads are obtained.

A series of theories has been developed and applied to analyze the buckling behavior of plates. One of these theories is the classical plate theory (CPT), described in study by Reddy [39]. It is used in the analysis of thin plates, but results in an underestimation of deflection and an overestimation of buckling loads. These theories are often called Refined Plate Theories (RPTs). They consist of first-order shear deformation theory (FSDT) and higher-order shear deformation theory (HSDT) (Onyeka et al. [40], Sayyad and Ghugal [41], Onyeka et al. [42]). Gunjal et al. [43] applied the improved trigonometric theory of plate shear deformation to analyze the uniaxial and biaxial compression behavior of a supported plate. For this purpose, they used the virtual work principle.

The polynomial shape functions were used by Ezeh et al. [44] and Onyeka et al. [45]. They analyzed the buckling behavior of the thick plate, simply supported and loaded with uniaxial compressive loading. In this case, polynomial shape theory was used to derive the plate equations. Ibeabuchi et al. [46] analyzed the buckling behavior of the plate using work principle approach. To obtain the buckling coefficients the polynomial displacement function was used, and finally, a mathematical model was created. However, the assumptions are limited to the classical plate theory and therefore are not reliable for thick plates.

Onyeka et al. [47] applied both trigonometric and polynomial displacement function to determine the critical buckling load of clamped thick rectangular plate using the analytical three-dimensional plate theory. This theory is formulated and derived from the variational energy method.

The main goal of this work is to test the different states of equilibrium of an object loaded with compressive forces and self-weight. Such structure may be a flat textiles (e.g., fabric) resting on a flat surface and supported on both ends. The obtained results can be used in algorithms and simulation programs for fabric folding, buckling and for another applications in the area of mechanics.

## 2. Materials

In this work, a flat strip of fabric is presented in the form of its longitudinal section. The numerical model presents a flat deflection curve, i.e., elastica. It has been shown in Figure 1. The particular longitudinal sections do not act on each other by internal forces. Furthermore, the constancy of properties along the whole width of the bending strip is assumed. Therefore, this study is limited to the analysis of the bending of heavy elastica. The bending stiffness of elastica is equal to *E*·*I*, and its self-weight per unit length is equal to *q*. The bending stiffness is equivalent to the product of modulus of elasticity times the moment of inertia of the cross-section.

## 3. Methods

### 3.1. Model Assumptions

The longitudinal section of the fabric of length *l* is loaded with self-weight *q* and compressive force *P*, as shown in Figure 2.

The structure rests on the flat surface and is fixed-hinged at points A and B. Of course, Hooke’s law applies in this case, and we can use the classical relation between the bending moment and curvature, M=EI/ρ. In this relationship, *ρ* is the radius of curvature. The rigid surface on which the tested object rests affects the constraints on the value of the y coordinate. For each coordinate *s* measured along the curve, *y* ≥ 0 must be satisfied. The following boundary conditions apply to this model.
$${\left.y(s)=0\right|}_{s=0}\;,\;{\left.y(s)=0\right|}_{s=l}\;,\;{\left.M(s)=0\right|}_{s=0}\;,\;{\left.M(s)=0\right|}_{s=l}$$

The infinitesimal section of the elastica is presented in Figure 3.

From the inextensibility condition, we have $dx^{2}/ds^{2}+dy^{2}/ds^{2}=1$. This leads to a geometrical condition of the form dy/ds<1. Next, we multiply the elementary equilibrium conditions by the virtual displacements δ*x*, δ*y* and δ*φ*. After adding the sides of the equations and then integrating from 0 to *l*, we have
(1)$$\begin{array}{c} -\int_{0}^{l}\frac{dP}{ds}\delta xds-\int_{0}^{l}\frac{dT}{ds}\delta yds-\int_{0}^{l}T\frac{dx}{ds}\delta \varphi ds\\ +\int_{0}^{l}P\frac{dy}{ds}\delta \varphi ds-\int_{0}^{l}q\delta yds+\int_{0}^{l}\frac{dM}{ds}\delta \varphi ds=0 \end{array}$$

Using integration by parts and appropriate boundary conditions, we obtain
(2)$$\int_{0}^{l}q\delta yds+P\delta x_{B}+\int_{0}^{l}\frac{1}{2}EI\delta {\left(\frac{d\varphi }{ds}\right)}^{2}ds=0$$

Equation (2) presents the principle of virtual work. It can be written in the following form:(3)δJ[y]=0
where
(4)$$J[y]=q\int_{0}^{l}yds+Px_{B}+\frac{1}{2}EI\int_{0}^{l}{\left(\frac{d\varphi }{ds}\right)}^{2}ds$$

The functional *J*[*y*] represents the total potential energy of the system. The necessary conditions for the extremum of functional *J*[*y*] are presented in Equation (3). It is known that the potential energy has a minimum value at the balance point in the case of stable equilibrium (stability). If the equilibrium is unstable, then the potential energy reaches its maximum value (labile equilibrium).

### 3.2. The Assumptions Regarding the Deflection Curve

For solving the problem of minimizing potential energy, the Rayleigh–Ritz method will be used. Let us assume that the deflection curve is given in the following form:(5)$$y(s)=A_{1}\sin\left(\pi s/l\right)+A_{2}\sin\left(3\pi s/l\right)$$

In this case, *A*_1_ and *A*_2_ represent the so-called shape parameters, describing different forms of bending curves, as shown in Figure 4.

Of course, the boundary conditions of the examined problem are met for Equation (5) because
$$\begin{array}{c} y\left(0\right)=y\left(l\right)=0,\;\mathrm{and}\\ M\left(s\right)=\frac{EI}{\rho (s)}=EI\cdot \kappa \left(s\right),\;\kappa \left(s\right)=1/\rho \left(s\right)=\frac{y^{\prime \prime }(s)}{{\left[1+y^{\prime 2}(s)\right]}^{3/2}},\;\kappa \left(0\right)=\kappa \left(l\right)=0 \end{array}$$

According to the Rayleigh–Ritz method, we obtain the functional *J*[*y*] as a function of two variables, *A*_1_ and *A*_2_*,* where *J*[*y*] = *V*(*A*_1_, *A*_2_). In order to find the values of the coefficients *A*_1_ and *A*_2_, we will use the necessary conditions of the extremum of the function *V*.
(6)$$\partial V/\partial A_{1}=0,\;\;\;\;\;\;\partial V/\partial A_{2}=0$$

Using Equation (5) in Equation (4), after some simplifications and the use of boundary conditions, we obtain the following relationship for potential energy.
(7)$$\begin{array}{c} V=Pl-P\frac{\pi^{2}}{4l}\left(A_{1}^{2}+9A_{2}^{2}\right)+\frac{2ql}{3\pi }\left(3A_{1}+A_{2}\right)\\ +\frac{\pi^{4}EI}{16l^{5}}\left(4A_{1}^{2}l^{2}+\pi^{2}A_{1}^{4}+12\pi^{2}A_{1}^{3}A_{2}+324A_{2}^{2}l^{2}+180\pi^{2}A_{1}^{2}A_{2}^{2}+729\pi^{2}A_{2}^{4}\right). \end{array}$$

In addition to the two variables *A*_1_ and *A*_2_, the energy *V*(*A*_1_, *A*_2_) is a function of two constants: *p* and *q*.

## 4. Results and Discussion

The values of variables *A*_1_ and *A*_2_ can be found using the necessary conditions of equilibrium. So let us apply the energy equation *V* (Equation (7)) to the conditions of the extremum (Equation (6)). For further considerations, let us assume the values of parameters *A*_1_ and *A*_2_, force *P,* and self-weight *q* in a dimensionless form using the following transformations:$$a=A_{1}/l,b=A_{2}/l,p=P/P_{kr},w=Q/P_{kr}$$
where Q=q⋅g, and $P_{kr}=\pi^{2}EI/l^{2}$ is Euler’s critical force.

Thus, we obtain
(8)$$\left\{\begin{array}{c} p=\frac{\pi^{2}}{2}a^{2}+\frac{9\pi^{2}}{2}ab+45\pi^{2}b^{2}+\frac{4w}{\pi^{3}}\cdot \frac{1}{a}+1\\ p=\frac{81\pi^{2}}{2}b^{2}+\frac{\pi^{2}}{6}\cdot \frac{a^{3}}{b}+5\pi^{2}a^{2}+\frac{4w}{27\pi^{3}}\cdot \frac{1}{b}+9 \end{array}\right.$$

These equations should be satisfied simultaneously. After first eliminating *p* and then *w* from Equation (8), we obtain the following relationships:(9)$$w\left(a,b\right)=\pi^{2}a^{4}+27\pi^{2}\left(a^{3}b-a^{2}b^{2}-ab^{3}\right)+48ab+\frac{24w}{27\pi^{3}}\left(a-27b\right)=0$$
(10)$$\begin{array}{c} p\left(a,b\right)=p\left(a-27b\right)-a+243b+\pi^{2}\left(4a^{3}-45ab^{2}\right)\\ +\frac{9}{2}\pi^{2}\left(243b^{3}+29a^{2}b\right)=0 \end{array}$$

Equation (9) presents the self-weight function *w*(*a*,*b*) in the so-called implicit form. Figure 5 presents the graph of function *w*(*a*,*b*). In the same way, the function *p*(*a*,*b*) is a function of two variables *a* and *b*. Its graph is shown in Figure 6.

The elastica is simultaneously loaded with self-weight *w* and force *p*. If we superimpose these both graphs over each other, we obtain the point of equilibrium at the intersection points of contour lines *p* and *w*. At these points, we have the extremum of the functional *J*[*y*]. Since Equation (6) is satisfied at these points, we expect a state of equilibrium there. Figure 7 presents the point of equilibrium for *w* = 12 and *p* = 10.

The next section of the work presents the analysis of the equilibrium states of the tested object.

### 4.1. Numerical Study of Equilibrium Points

As stated above, the energy *V* is a function of two variables *a* and *b*. In this case, equilibrium analysis is nothing more than searching for the extremum of two variables function. The extremum depends upon the following two conditions:
–Sign of second derivative $\partial^{2}V/\partial a^{2}$;–Value of discriminant *W*, where $W=\left(\frac{\partial^{2}V}{{\partial a}^{2}}\right)\cdot \left(\frac{\partial^{2}V}{{\partial b}^{2}}\right)-{\left(\frac{\partial^{2}V}{\partial a\;\partial b}\right)}^{2}$.If the necessary conditions in Equation (6) are met and *W* > 0, then:–If $\partial^{2}V/\partial a^{2}>0$, we have the state of stable equilibrium at this point (the energy function *V* reaches a minimum value);–If $\partial^{2}V/\partial a^{2}<0$, we have the state of labile equilibrium at this point (the energy function *V* reaches a maximum value).


Taking into account Equation (7), it is possible to find the discriminant *W* and appropriate derivatives. Moreover, to have dimensionless values, a slight transformation is necessary. It means that *W* must be divided by $l^{2}/P_{kr}^{2}$, and that $\partial^{2}V/\partial a^{2}$ must be divided by $l/P_{kr}$.

We obtain the following relationships:(11)$$W\left(l^{2}/P_{kr}^{2}\right)=\left[\frac{\pi^{2}}{2}\left(1-p\right)+\frac{3\pi^{4}}{4}\left(a^{2}+6ab+30b^{2}\right)\right]\;\left[\frac{9\pi^{2}}{2}\left(9-p\right)+\frac{9\pi^{4}}{4}\left(243b^{2}+10a^{2}\right)\right]-{\left[\frac{9\pi^{4}}{4}\left(a^{2}+20ab\right)\right]}^{2}$$
(12)$$(\partial^{2}V/\partial a^{2})(l/P_{kr})=\frac{\pi^{2}}{2}\left(1-p\right)+\frac{3\pi^{4}}{4}\left(a^{2}+6ab+30b^{2}\right)$$

From the above equations, we can see that the value of the second derivative *V* depends only on the force *p* and parameters *a* and *b*, and this is likewise for the value of discriminant *W*. To change the type of equilibrium of the system, it is enough to change the *p* value.

The sign of the second derivative $\partial^{2}V/\partial a^{2}$ and the value of the discriminant *W* have been found numerically. As a result of a numerical analysis, certain areas were obtained in which various forms of equilibrium occur. There is a stable equilibrium in one area and an unstable equilibrium in another. There are also areas where the equilibrium does not exist (*W* ≤ 0). Examples of equilibrium areas are shown in Figure 8 and Figure 9.

From these figures, it can be concluded that

–For *p* = 1.5 in Figure 8, in the dark area, only a stable equilibrium exists, while in the light area, the equilibrium does not exist.–For *p* = 12 in Figure 9, in the dark area, the labile equilibrium exists.

### 4.2. The Shape Parameters and Their Permissible Values

The values of the shape parameters *A*_1_ and *A*_2_ (and their dimensionless forms *a* and *b*) in Equation (5) cannot take arbitrary values. So let us consider how to determine their acceptable values. A rigid surface, as shown in Figure 2, leads to the following geometric condition: *y* ≥ 0 for 0 ≤ *s* ≤ *l*. Moreover, inextensibility implies that *dy*/*ds* < 1 for 0 ≤ *s* ≤ *l*. Let us further define a dimensionless variable: ξ=πs/l. According to Equation (5), we have that
(13)$$\begin{array}{c} a\sin\left(\xi \right)+b\sin\left(3\xi \right)\geq 0\\ \pi a\cos\left(\xi \right)+3\pi b\cos\left(3\xi \right)<1,\;\;\;\;\;\;for\;0\leq \xi \leq \pi \end{array}$$

From the entire range of values of *a* and *b*, those that satisfy the inequalities (13) should be selected. The inequalities (13) have been calculated numerically. The extreme values of parameters *a* and *b* are presented below.
$$a_{min}=0.0,\;a_{max}=0.3676,\;b_{min}=-0.0707,\;b_{max}=0.0796$$

Admissible area of *a* and *b* are shown in Figure 10.

Testing of equilibrium, the values of parameters *a* and *b* within the permissible range should be taken into account (Figure 10). For this purpose, the admissible values of *a* and *b* are superimposed on the graph in Figure 8. 

Concluding the analysis, it should be added that the above considerations are a generalization of the problem discussed in [15], where the deflection curve is dependent on one parameter only.

## 5. Conclusions

In this article, an energy method for analyzing a heavy elastica loaded with compressive force and self-weight has been presented. A test of equilibrium states was carried out for a given deflection curve, which depended on two shape parameters. By changing the value of shape parameters *A*_1_ and *A*_2_, it is possible to simulate different shapes of the deflection curve. Therefore, the analysis of equilibrium states can be performed for various shapes of the bending curve. To solve the presented problem, the Rayleigh–Ritz energy method was used. This considerations is a generalization of the issue discussed in the author’s earlier work where the deflection curve was dependent only on one parameter. The presented problem was non-linear, so numerical methods were used to solve it. It is possible to generalize the problem to a multidimensional task, where there are more than two shape parameters. The obtained results may be used in algorithms and simulation programs for fabric folding, buckling and for other applications in the area of mechanics.

## Figures and Tables

**Figure 1 materials-17-00503-f001:**
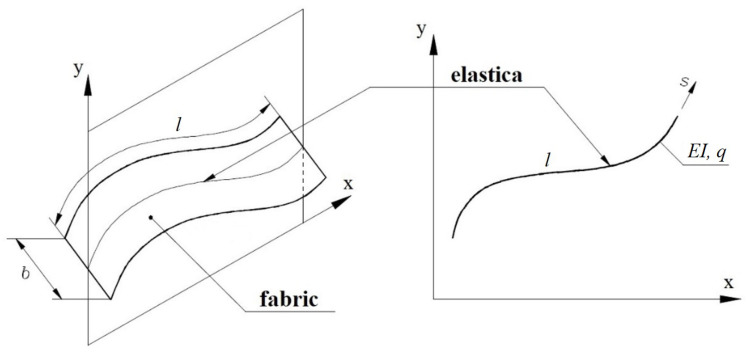
The model of fabric approximated by elastica.

**Figure 2 materials-17-00503-f002:**
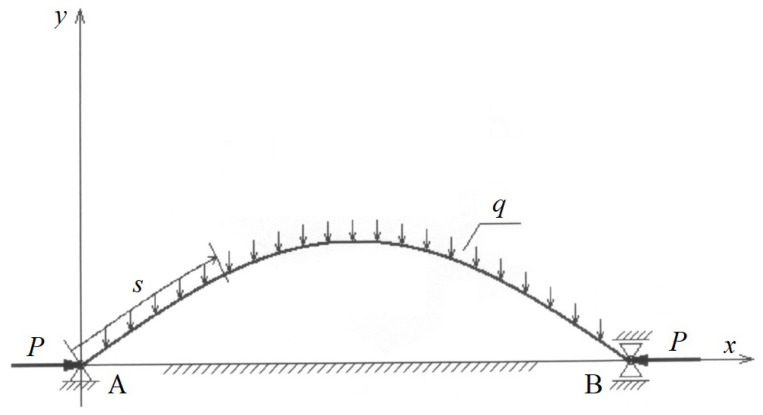
Elastica loaded with compressive forces.

**Figure 3 materials-17-00503-f003:**
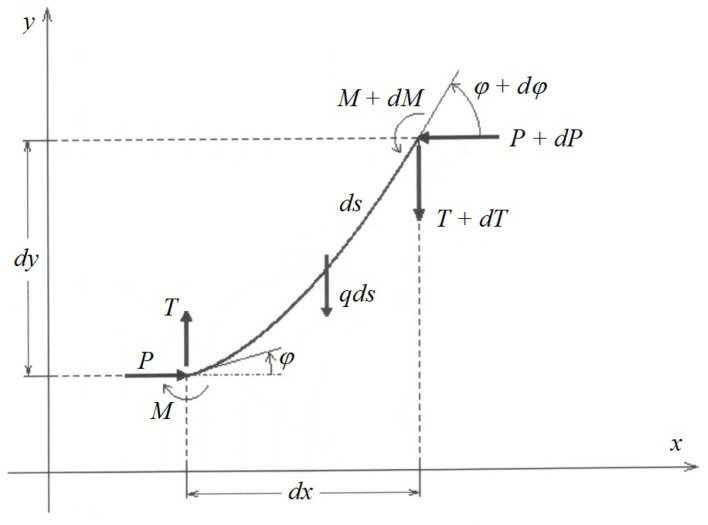
Forces acting on an infinitesimal section of elastica.

**Figure 4 materials-17-00503-f004:**
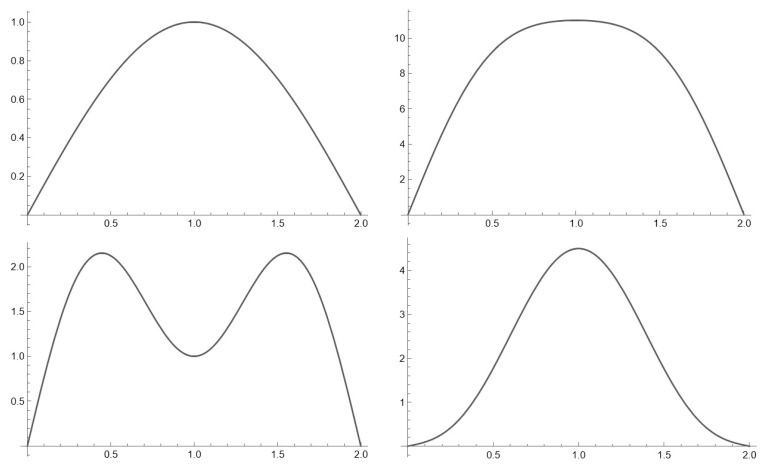
The different shapes of the deflection curve according to Equation (5).

**Figure 5 materials-17-00503-f005:**
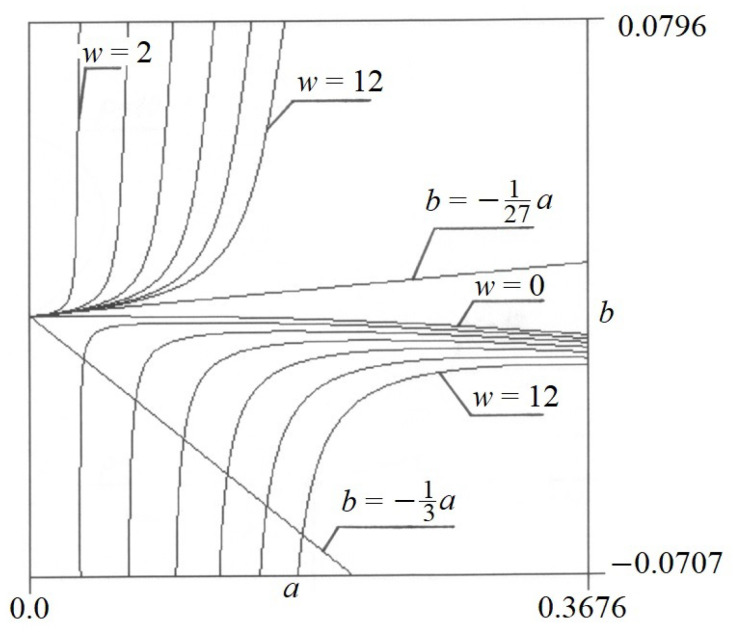
The graph of function *w*(*a*,*b*).

**Figure 6 materials-17-00503-f006:**
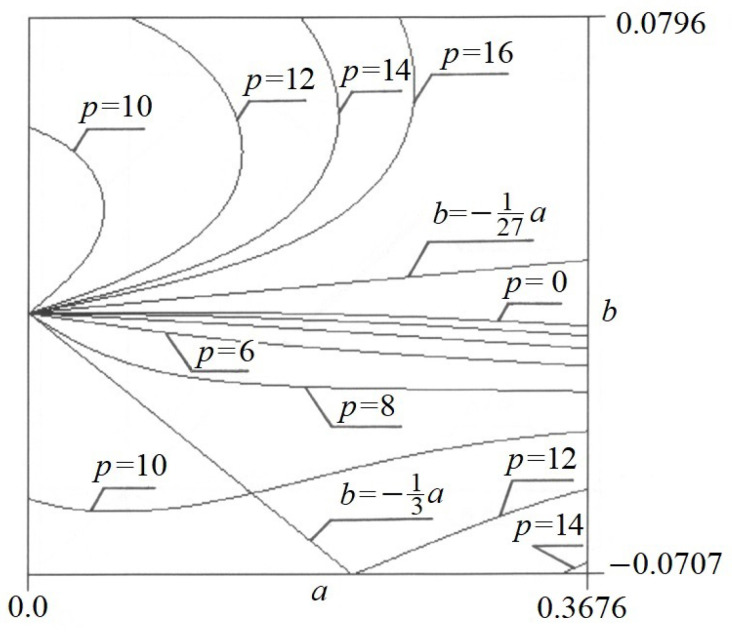
The graph of function *p*(*a*,*b*).

**Figure 7 materials-17-00503-f007:**
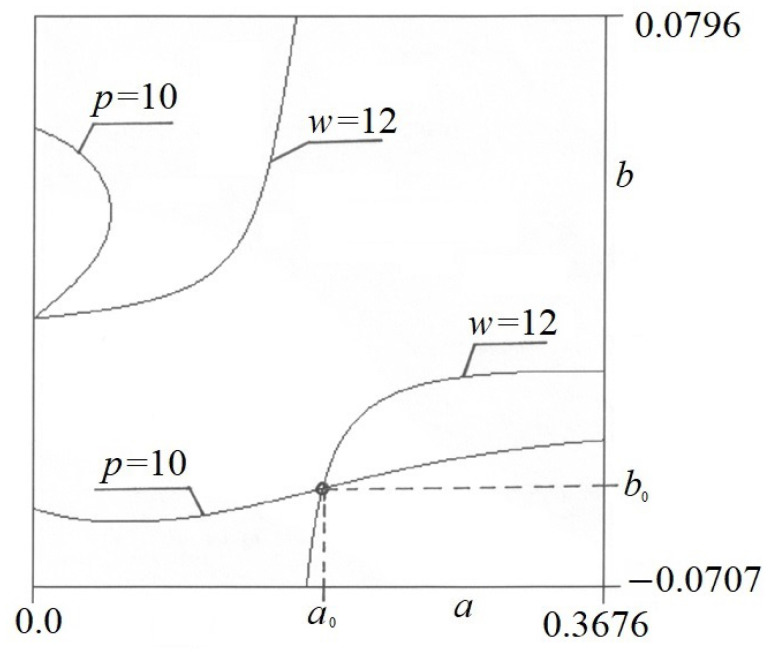
The sample point of state of equilibrium for *p* = 10 and *w* = 12.

**Figure 8 materials-17-00503-f008:**
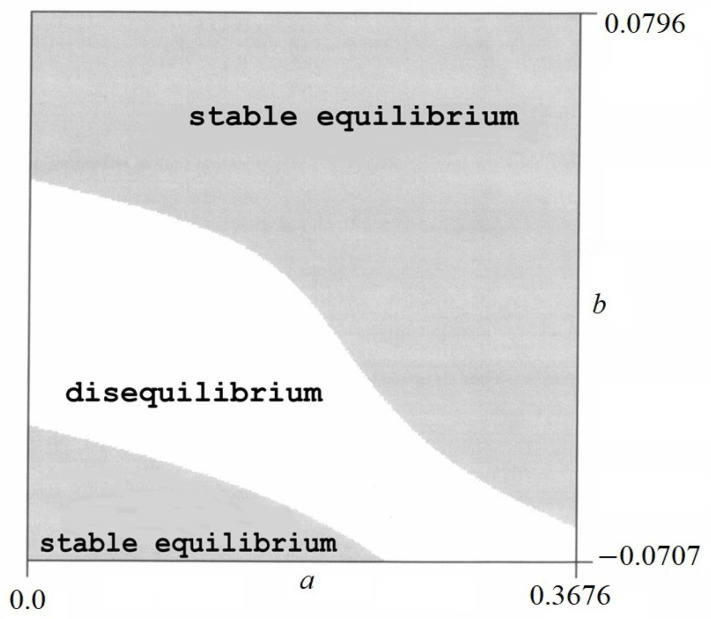
Areas of equilibrium states for *p* = 1.5.

**Figure 9 materials-17-00503-f009:**
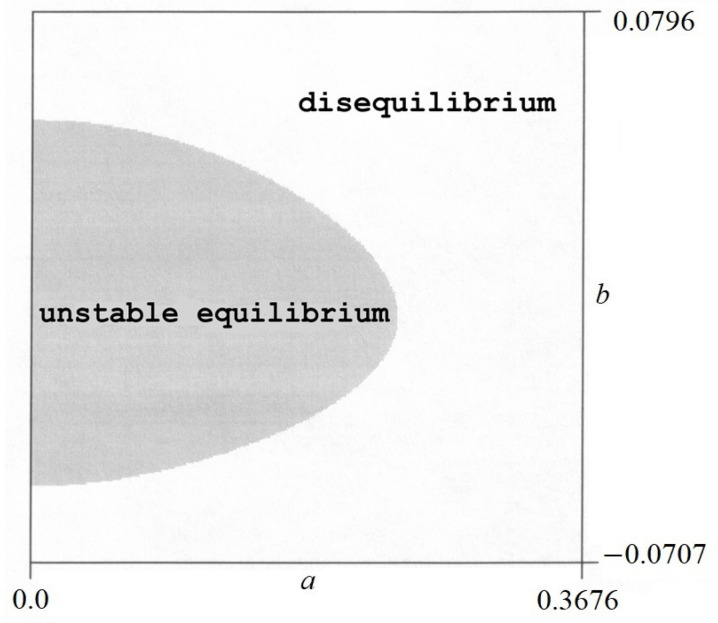
Areas of equilibrium states for *p* = 12.

**Figure 10 materials-17-00503-f010:**
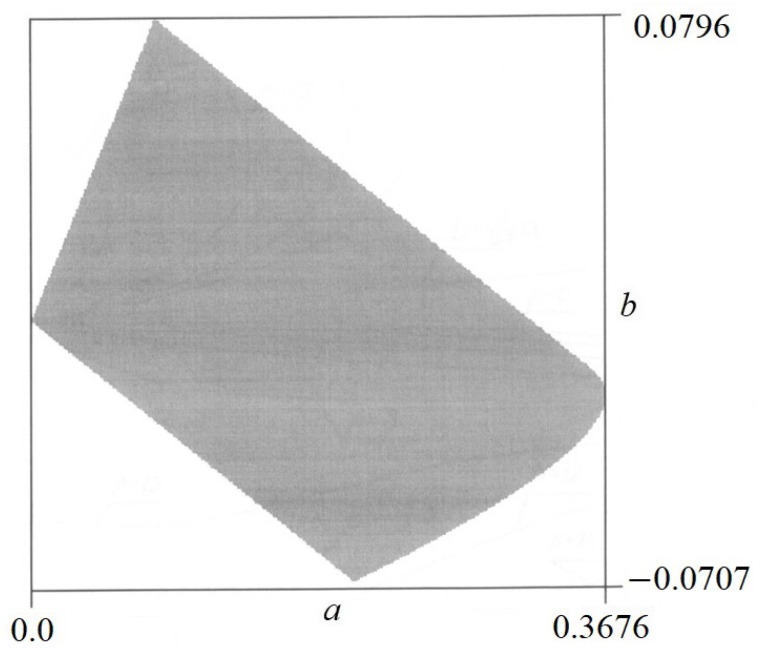
Admissible area of parameters *a* and *b*.

## Data Availability

The data presented in this study are available on request from the corresponding author.

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
