# Peer review of "Analysis of the Stability of a Flat Textiles with a Two-Parameter Deflection Curve"

_materials, 2024, doi:10.3390/ma17020503_

Round 1

Reviewer 1 Report

Comments and Suggestions for Authors

The authors tested the equilibrium states of a flat textiles loaded by self-weight and compressive forces. By changing the value of the two shape parameters, it is possible to simulate different shapes of the deflection curve. Therefore, the analysis of equilibrium states can be performed for various shapes of the bending curve. After identifying several issues that require attention, I recommend a major revision of the manuscript.

1. I recommend revising the abstract to include a concise summary of the research significance.

2. Introduction's logic is confusing. Why is it necessary to study deflection issues in textile mechanics? Under what conditions do textiles exhibit large deflection? The research significance of the paper is not clearly explained.

3. The references in the introduction that elaborate on deflection are all related to plates or beams. Are there any references specifically addressing deflection in textile fabrics?

4. What improvements does the author make on the basis of previous research, and where does the novelty of the paper lie? How does the author adapt the deflection curve to textile mechanics based on previous work?

5. For simulation analysis, if conditions permit, there should be corresponding experimental part, so as to form a complete research closed-loop.

Comments on the Quality of English Language

 Minor editing of English language required

Author Response

Thank you very much for taking the time to review this manuscript. The comments you indicate will contribute to the improvement of the author's future works. Thank You very much.

A new version of the revised manuscript has been submitted. New or corrected content is marked in red in the manuscript (materials-2782690-corrected.docx).

1. The abstract has been revised. New content regarding textile mechanics has been inserted.

2. The issue of large deflections of textiles occurs primarily when analyzing the “drape” of curtains, tablecloths and other flat textile products. Correct drape is particularly important from an aesthetic point of view. Additionally, there is a problem with the stability of the folds created in the drape process.
Large deflections also occur when folding flat textiles (a very important technological process).
In this article, a selected fragment of the above-mentioned issue was tested, relating only to the stability of the fold formed under given loading conditions.

3. Due to the fact that the textile mechanics is a relatively young field of science, many problems are solved on the basis of classical mechanics, the theory of elasticity and mechanics of beams, plates and shells. Let me give you one example. The work:
Lloyd, D.W. The Mechanics of Drape. In Flexible Shells. Theory and Applications, 1st ed.; Axelrad, E.L., Emmerling, F. A., Eds.; Springer-Verlag: Berlin, Germany, 1984, pp. 271–282.
Lloyd considered the drape process of textile structures. In this work the special difficulties of modelling the complex deformations of textile fabrics are discussed in the context of flexible shell theory, and the particular requirements of analysis methods in this area are described.
Major changes have been made to the references section. Literature was supplemented with publications from the area of textile mechanics.

There are references specifically addressing deflection in textile fabrics. See below:
Ref. [1], [2], [3], [4], [5], [6], [7], [8], [15], [22], [26].

4. To the author's knowledge, no similar study has been found in the professional literature. The stability analysis of the two-parameter deflection curve is the author's original idea. It should be noted that the presented work is a generalization of the problem discussed at author's previous works.

5. The work is at the development stage, so the author was not able to perform appropriate experiments at the moment. These are theoretical considerations that can be verified virtually (apart from the experiment) using finite element method software.

Thank you once again for your review and I would like to ask You to respond positively to my work.

Reviewer 2 Report

Comments and Suggestions for Authors

The comments listed below points out some of the main problems, and hope these suggestions can be helpful for the authors to improve their work.

1 Language expression. Please polish the language and check for grammatical errors to make the sentence more readable and logical. From the perspectives of both the novelty of this work and the writing of this manuscript, the innovation of this manuscript needs to be further emphasized.

2 This is methods of model for fabric approximated by elastica, no relying on the experimental results. The significance of the practical application and guidance of this research result should be discussed, and strong factual support should be provided.

3 SECTION of conclusions, it should be revised according to the requirement of journal, please revise the format of the conclusion section. What is more, the references need to add recently three years.

Comments on the Quality of English Language

The comments listed below points out some of the main problems, and hope these suggestions can be helpful for the authors to improve their work.

1 Language expression. Please polish the language and check for grammatical errors to make the sentence more readable and logical. From the perspectives of both the novelty of this work and the writing of this manuscript, the innovation of this manuscript needs to be further emphasized.

2 This is methods of model for fabric approximated by elastica, no relying on the experimental results. The significance of the practical application and guidance of this research result should be discussed, and strong factual support should be provided.

3 SECTION of conclusions, it should be revised according to the requirement of journal, please revise the format of the conclusion section. What is more, the references need to add recently three years.

Author Response

Thank you very much for taking the time to review this manuscript. The comments you indicate will contribute to the improvement of the author's future works. Thank You very much.

A new version of the revised manuscript has been submitted. New or corrected content is marked in red in the manuscript (materials-2782690-corrected.docx).

The Abstract and Introduction have been revised. New content regarding textile mechanics has been inserted.

To the author's knowledge, no similar study has been found in the professional literature. The stability analysis of the two-parameter deflection curve is the author's original idea. It should be noted that the presented work is a generalization of the problem discussed at author's previous works.

The work is at the development stage, so the author was not able to perform appropriate experiments at the moment. These are theoretical considerations that can be verified virtually (apart from the experiment) using finite element method software.

Major changes have been made to the references section. Literature was supplemented with publications from the area of textile mechanics.

More of the discussion takes place in the Results section. Therefore, its name was changed to: Results and Discussion.

The Reference list includes titles from the last three years.

The manuscript was reviewed by an experienced English-speaking colleague who concluded that the text did not require major changes and is acceptable.

Thank you once again for your review and I would like to ask You to respond positively to my work.

Reviewer 3 Report

Comments and Suggestions for Authors

The author present the paper entitled: "Analysis of the Stability of a Flat Textiles with a Two-Parameter Deflection Curve". Having reviewed it point by point, the comments made are set out below:

1.   Abstract: The way the abstract is written does not make it clear what the fundamental objective of this work is. Nor is the methodology applied in each phase of the research clearly stated. The author should begin the abstract with a brief introduction in which the field of action is contextualised.

2. Introduction: the Introduction is extremely long (almost three pages) considering that the paper is 11 pages in total. Therefore, there is a notable imbalance with respect to the methodological part of the work. It is recommended that the Introduction be made more concise.

3. Figures: All figures need higher resolution and larger size so that all details can be identified.

4. The "Discussion of results" section is missing. It is recommended that the author add this new section in which he/she can discuss, interpret and compare the results obtained with those of previous researchers. For example, in the Introduction, reference is made to 43 bibliographical citations, however, no mention is made of any of these authors during the description of the results.

5.Conclusions: conclusions need to be rewritten. The arguments put forward are too general and confusing for readers.

6. Conclusions: in this section, formulas (see Line 326), figures (see Line 328) and other authors (see Line 331) should not be cited. All such citations should be moved to the "Discussion of results" section

Author Response

Thank you very much for taking the time to review this manuscript. The comments you indicate will contribute to the improvement of the author's future works. Thank You very much.

A new version of the revised manuscript has been submitted. New or corrected content is marked in red in the manuscript (materials-2782690-corrected.docx).

- The Abstract and Introduction have been revised. New content regarding textile mechanics has been inserted.

- I am very sorry for the too long introduction. I wanted to discuss in depth the state of knowledge regarding the issues discussed in this work. Perhaps too broadly discussed. I will definitely take this into account in future work.

- Regarding drawings - if necessary, the technical editor will increase the resolution for better quality.

- I have included most of the discussion in the Results section. Therefore, I changed the name of this section to: Results and Discussion.
- The bibliographic citations were intended to inform the reader about the state of knowledge regarding large deflections and stability issues. In the discussion of the results, the author refers particularly to work [15], which is the starting point for this article.

- The Conclusion section has been revised.
- Citations to formulas, drawings and other authors have been removed.

Thank you once again for your review and I would like to ask You to respond positively to my work.

Reviewer 4 Report

Comments and Suggestions for Authors

The paper delves into the analysis of equilibrium states for a flat textile structure subjected to self-weight and compressive forces. Focused on fabric resting on a surface with support at both ends, the study outlines the deflection curve generation for varying self-weight and compressive forces. Two shape parameters facilitate the simulation of diverse deflection curve shapes. Employing the energy method, the analysis examines the total potential energy of the structure. The results offer potential applications in algorithms for fabric folding, buckling, and broader mechanical fields.

My Comments:

1. How was the energy method applied precisely to analyze the total potential energy of the textile structure under self-weight and compressive forces?

2. How sensitive are the shape parameters A1 and A2 in generating different deflection curve shapes? Are there other parameters influencing these curves?

3. Besides fabric folding and buckling, what specific algorithms or simulation programs in mechanical applications could directly benefit from these results?

4. Could the outcomes obtained from this energy method be compared or validated against other numerical methods, particularly those applied in textile mechanics?

5. Were there any inherent limitations or assumptions in the Rayleigh-Ritz energy method applied? Were these limitations tested or addressed in the context of practical textile mechanics?

6. Beyond fabric-related applications, are there potential cross-disciplinary applications or industries where these findings might have implications?

7. What potential future research pathways or extensions do you envision based on the findings obtained from this study, especially concerning the analysis of flat textile structures under varying conditions?

Author Response

Thank you very much for taking the time to review this manuscript. The comments you indicate will contribute to the improvement of the author's future works. Thank You very much.

A new version of the revised manuscript has been submitted. New or corrected content is marked in red in the manuscript (materials-2782690-corrected.docx).

1) First of all, a relationship for the total potential energy of the tested system was derived.
From Figure 3, the equilibrium conditions are as follows. They are multiplied by the appropriate virtual displacements.
-dP=0, -dT-qds=0, dM-Tdx+Pdy=0.
After adding the sides of the equations and integrating from 0 to l, we have Equation 1.
After integrating by parts we get:

Using:

and appropriate boundary conditions:

we get Equation 2, next Equation 3 and 4. Equation 2 presents the principle of virtual work.
The functional 4 represents the total potential energy of the system. Such total potential energy for various  load conditions can be found in classical mechanics and theory of elasticity.
Therefore, the total potential energy results directly from the principle of virtual work.

The extremes of potential energy were then examined to determine the type of equilibrium.
The energy method in this case means using potential energy to solve the problem. 

2) Thank you for this question. Unfortunately, at this stage of work, the influence of shape parameters on  generating different deflection curve shapes was not examined. This is a good suggestion.  It may be taken into account in the future development of this issue.`

The influence of other parameters was not taken into account in this work.

3) The presented solution can be used, for example, to numerical determine deflection curves of linear structures, cantilever beams, cables and ropes. The method can be implemented in software. But this was not the purpose of this work.

4) The author did not verify the results in this work. The considerations were purely theoretical. But verification is possible. For example, using finite element software. For this purpose, you would need to create a model in applications such as SolidWorks, Solid Edge, Ansys, etc.
Such software is an excellent virtual laboratory. This suggestion is the next starting point for developing this issue.

5) Only the permissible values of the shape parameters as a limitation in the method were tested. See point 4.2. No other limitations or assumptions were taken into account.

6) Of course, the results of the work can be used in the analysis of various mechanical structures. Everything depends only on the appropriate selection of parameters and mechanical properties.
The presented solution can be used to numerical determine deflection curves of linear structures, cantilever beams, cables and ropes.

7) Future research pathways or extensions:
- verification of the results using FEM software,
- testing the sensitivity of shape parameters to the obtained results,
- taking into account the type of weave or variable cross-section in the fabric,
- considering other types of loading.

Thank you once again for your review and I would like to ask You to respond positively to my work.

Round 2

Reviewer 2 Report

Comments and Suggestions for Authors

The structure and logic of this article is relatively complete, according to the suggestion before, the authors have revised this article including the language, the authors also gave the reasonable expression about the questions. But it should be suggested that the authors go on to revise grammar and format, interpunction.

Comments on the Quality of English Language

The structure and logic of this article is relatively complete, according to the suggestion before, the authors have revised this article including the language, the authors also gave the reasonable expression about the questions. But it should be suggested that the authors go on to revise grammar and format, interpunction.

Reviewer 3 Report

Comments and Suggestions for Authors

The author has revised and improved the work according to the suggestions and indications of the reviewer.

Reviewer 4 Report

Comments and Suggestions for Authors

Accept in present form.